# Outlier Detection from Image Data

## Abstract

Modern applications from Autonomous Vehicles to Video Surveillance generate massive amounts of image data. In this work we propose a novel image outlier detection approach (IOD for short) that leverages the cutting-edge image classifier to discover outliers without using any labeled outlier. We observe that although intuitively the confidence that a convolutional neural network (CNN) has that an image belongs to a particular class could serve as outlierness measure to each image, directly applying this confidence to detect outlier does not work well. This is because CNN often has *high confidence* on an outlier image that does not belong to any target class due to its generalization ability that ensures the high accuracy in classification. To solve this issue, we propose a Deep Neural Forest-based approach that harmonizes the contradictory requirements of accurately classifying images and correctly detecting the outlier images. Our experiments using several benchmark image datasets including MNIST, CIFAR-10, CIFAR-100, and SVHN demonstrate the effectiveness of our IOD approach for outlier detection, capturing more than 90% of outliers generated by injecting one image dataset into another, while still preserving the classification accuracy of the multi-class classification problem.

## 1 Introduction

**Motivation.** As modern applications such as autonomous vehicles and video surveillance generate larger amount of image data, the discovery of outliers from such image data is becoming increasingly critical. Examples of such image outliers include unauthorized personnel observed in a secret military base or unexpected objects encountered by self-driving cars on the road. Capturing these outliers can prevent intelligence leaks or save human lives.

**State-of-the-Art.** Due to the exceptional success of deep learning over classical methods in computer vision, in recent years a number of works (Makhzani & Frey, 2015; Schlegl et al., 2017; Erfani et al., 2016; Perera & Patel, 2018) leverage the representation learning ability of a deep autoencoder or GAN (Goodfellow et al., 2016) for outlier detection. Outliers are either detected by plugging in the learned representation into classical outlier detection methods or directly reported by employing the reconstruction error as the outlier score (Zhou & Paffenroth, 2017; Chen et al., 2017). However, these approaches use a generic network that is not trained specifically for outlier detection. Although the produced representation is perhaps effective in representing the common features of the "normal" data, it is not necessarily effective in distinguishing "outliers" from "inliers". Recently, some works (Ruff et al., 2018; Nguyen & Vien, 2018) were proposed to solve this issue by incorporating the outlier detection objective actively into the learning process. However, these approaches are all based on the one-class technique (Schölkopf et al., 2001; Manevitz & Yousef, 2002; Tax & Duin, 2004) that learns a single boundary between outliers and inliers. Although they perform relatively well when handling simplistic data sets such as MNIST, they perform poorly at supporting complex data sets with multiple "normal" classes such as CIFAR-10 (Krizhevsky & Hinton, 2009). This is due to the difficulty in finding a separator that encompasses all normal classes yet none of the outliers.

**Proposed Approach and Contributions.** In this work we propose a novel image outlier detection (IOD) strategy that successfully detects image outliers from complex real data sets with multiple normal classes. IOD unifies the core principles of cutting edge deep learning

image classifiers (Goodfellow et al., 2016) and classical outlier detection (Aggarwal, 2017) within one framework.

Classical outlier detection techniques (Aggarwal, 2017; Breunig et al., 2000; Knorr & Ng, 1998; Bay & Schwabacher, 2003) consider an object as an outlier if its outlierness score is above a certain cutoff threshold $ct$. Intuitively given a Convolutional Neural Network (CNN) (Krizhevsky et al., 2012a) trained using normal training data (namely, data without labeled outliers), the confidence that the CNN has that an image belongs to a particular class could be leveraged to measure the outlierness of the image. This is based on the intuition that we expect a CNN to be less confident about an outlier compared to inlier objects, since outliers by definition are dissimilar from any normal class. By using the confidence as an outlier score, IOD could separate outliers from all normal classes.

However, our experiments (Sec. 2) show that directly using the confidence produced by CNN to identify outliers in fact is not particularly effective. This is because the requirements of accurately classifying images and correctly detecting the outlier images conflict with each other. CNN achieves high accuracy in image classification because of its excellent generalization capability that enables a CNN to overcome the gap between the training and testing images. However, the generalization capability jeopardizes the detection of outliers, because it increases the chance of erroneously assigning an outlier image to some class *with high confidence* to which actually it does not fit.

We solve this problem by proposing a deep neural decision forest-based (DNDF) approach equipped with an information theory-based regularization function that leverages the strong bias of the classification decisions made within each single decision tree and the ensemble nature of the overall decision forest. Further, we introduce a new architecture of the DNDF that ensures independence amongst the trees and in turn improves the classification accuracy. Finally, we use a joint optimization strategy to train both the spit and leaf nodes of each tree. This speeds up the convergence.

We demonstrate the effectiveness of our outlierness measure, the deep neural forest-based approach, the regularization function, and the new architecture using benchmark datasets including MNIST, CIFAR-10, CIFAR-100, and SVHN – with the accuracy higher than 0.9 at detecting outliers, while preserving the accuracy of multi-class classification.

## 2 Confidence-based Outlier Detection

In convolutional neural networks (CNN) (Krizhevsky et al., 2012b), the final Fully Connected (FC) layer contains a single node $n_i$ for each target class $C_i$ in the model. Here $i \in \{1, 2, ..., m\}$, where $m$ is the number of target classes. Given an input image, the final FC layer computes a weighted sum score, $s_i$ w.r.t. class $C_i$ as $s_i = \sum_{j=1}^{n} F_j w_{ji}$, where $F_j$ is a feature produced by the convolutional model and $w_{ji}$ is the learned weight of the FC layer that connects $F_j$ and node $n_i$. The *maximum weighted sum* is the largest weighted sum score among all classes, defined as $max(s_1, s_2, ..., s_m)$.

Using the weighted sum scores as input, a softmax activation function is then applied to generate a class probability $p_i$ between 0–1 for each node $n_i$ ($\sum_{i=1}^{m} p_i = 1$). This can be interpreted as relative measurement of how likely it is that one image falls into target class $C_i$. A testing image $t$ will be assigned to class $C_i$ if $p_i = max(p_1, p_2, ..., p_m)$. $p_i$ is called the *maximum probability* of image $t$.

Intuitively, if an image does not have a good match with any known class, its maximum probability will be small. Since outliers typically do not exhibit many features of a known class as compared to inliers, we postulate the maximum probability of an outlier tends to be small relative to inliers. Therefore, it is natural to use the *maximum probability* or the corresponding *maximum weighted sum score* as a confidence measure for images.

**Confidence-based Outlier Detection.** Next, we propose a method that uses this confidence to detect outliers. An outlierness threshold $ct$ is established in the training process using the training image $x_k$ which has the $k$th smallest confidence among all objects in the training set. Then given a testing image, it will be considered as an outlier if its confidence is

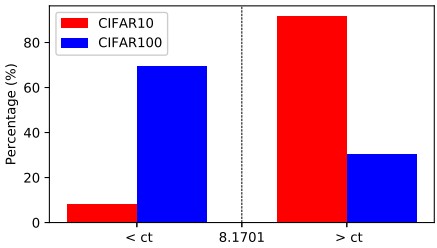 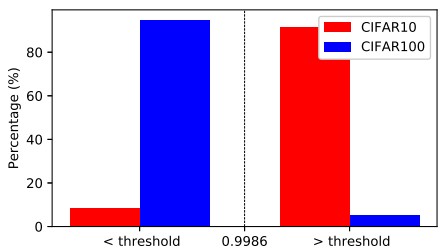

(a) Maximum weighted sum distribution

(b) Our improved score using max route probability distribution

Figure 1: Max weighted sum VS max route probability (tested on CIFAR-10 model).

smaller than the confidence of this selected $x_k$. One advantage of this approach is that unlike the state-of-the-art image outlier detection work (Ruff et al., 2018; Schlegl et al., 2017) we do not have to explicitly set the outlierness cutoff threshold $ct$ at the cost of introducing a different hyperparameter $k$. However, $k$ is a more intuitive parameter to set than $ct$, because $k$ can be set by simply assuming a percentage of the outliers in the training data.

Our experiments show that this straightforward approach alone does not work very well, because many of the outlier images do have a large maximum probability (maximum weighted sum). We illustrate this with an example. First, we trained a CIFAR-10 model using the CIFAR-10 dataset (Krizhevsky & Hinton, 2009). We then used the model to classify the CIFAR-10 and CIFAR-100 testing data. Each dataset contains 60,000 images. Ideally, all CIFAR-100 images should be detected as outliers, because the CIFAR-100 and CIFAR-10 images contain disjoint classes. Fig. 1(a) shows the maximum weighted sum w.r.t. the CIFAR-10 and CIFAR-100 datasets when tested on the CIFAR-10 model. The CNN network is identical to the network we described in our experiment section (Sec. 5.2). Note we demonstrate the results of using maximum weighted sum instead of maximum probability, because our experimental evaluation shows that maximum weighed sum performs bettern than maximum probability in detecting outliers. The black vertical line represents the cutoff threshold $ct$. In this case, $k$ is set as 5000. We expect nearly 100% of CIFAR-100 images to fall to the left of the black dashed line and hence be captured as outliers. However, more than 30% CIFAR-100 images have large maximum weighted sum scores and are not correctly classified as outliers. Thus, the accuracy of this outlier detection scheme is less than 0.7.

**Analysis.** The low accuracy of relying on the *maximum probability* or *maximum weighted sum* to detect outliers is caused by the contradictory requirements of accurately classifying images and effectively detecting outliers. It is well known that to accurately classify images, the image classifier has to have excellent generalization capability such that the gap between the training and testing images can be overcome – in other words avoiding overfitting. For this, regularization functions such as data augmentation (Simard et al., 2003), random dropout (Srivastava et al., 2014), or weight decay (Krogh & Hertz, 1991) are commonly used to improve the generalization of CNNs. However, these generalization methods inevitably jeopardize the outlier detection capability of the model. This is because some images will be classified to one class even if they are quite different from the common features of that class extracted during training. Yet instead such images might be outliers – generalization tends to blur the boundary between normal images and outliers. Therefore, high confidence may be assigned to image outliers as shown in above use case.

## 3   DEEP NEURAL DECISION FOREST-BASED APPROACH

To address this shortcoming of simply using the maximum class probability as the outlier score, we propose a deep neural decision forest-based approach that harmonizes the contradictory requirements of accurate image classification and effective outlier detection in one network structure. It does this by taking advantage of the strong bias of the classification decisions made within each tree and the ensemble nature of the decision forest. In this section, we

first introduce the deep neural forest model (Kontschieder et al., 2015). We then show how to use the *maximum route* of each tree to distinguish outliers from inliers.

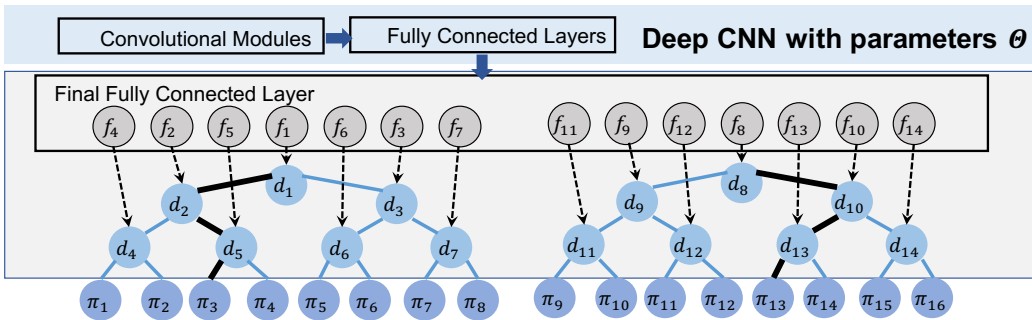

Figure 2: Deep Neural Forest Structure.

## 3.1 Deep Neural Decision Forest

The *deep neural forest* (Kontschieder et al., 2015) combines the representation learning of deep convolutional networks and the divide-and-conquer principle of decision trees. The key idea is to introduce a back-propagation compatible version of stochastic decision trees. As depicted in Fig. 2, the deep neural forest is composed of three key components: a deep CNN, decision nodes (split nodes), and prediction nodes (leaf nodes). The deep CNN component holds parameters from all convolutional modules and the FC layers except for the final FC layer. Decision nodes indexed by $\mathcal{N}$ are the internal nodes of the tree. Each decision node is connected to one output node of the final FC layer. Prediction nodes indexed by $\mathcal{L}$ are the leaf nodes of the tree.

Each decision node $n \in \mathcal{N}$ is assigned a decision function $d_n(.;\Theta) : \mathcal{X} \to [0,1]$ parametrized by $\Theta$. Each decision node $n$ classifies a sample $x$ based on its features and routes it left or right down the tree. Note that the routing decision $d_n$ is probabilistic; formally $d_n(x;\Theta)$ is defined as:

$$d_n(x;\Theta) = \sigma(f_n(x;\Theta)) \tag{1}$$

where $\sigma(x) = (1 + e^{-x})^{-1}$ is the sigmoid function and function $f_n(.;\Theta)$ is a linear output unit provided by the FC layer of a deep network parametrized by $\Theta$. Therefore, it is $f_n(.;\Theta)$ that connects the neural network and the decision tree. The function is turned into a probabilistic routing decision in the $[0,1]$ range by applying the sigmoid activation.

Each prediction node $l \in \mathcal{L}$ holds a probability distribution $\pi_l$ over the output space $\mathcal{Y}$. Once a sample ends at a leaf node $l$, a prediction is given by $\pi_l$. In the case of stochastic routing, the leaf predictions will be averaged by the probability of reaching the leaf.

The final prediction for sample $x$ from tree $T$ with decision nodes parametrized by $\Theta$ is given by

$$\mathbb{P}_T[y|x,\Theta,\pi] = \sum_{l \in \mathcal{L}} \pi_{ly} \mu_l(x|\Theta) \tag{2}$$

where $\pi = (\pi_l)_{l \in \mathcal{L}}$ and $\pi_{ly}$ denotes the probability that a sample reaching leaf $l$ belongs to class $y$. $\mu_l(x|\Theta)$ is considered as the *routing function*. It produces the probability that sample $x$ will reach leaf $l$. $\sum_l \mu_l(x|\Theta) = 1$ for all $x \in \mathcal{X}$. Since $\pi_l$ is not influenced by the the input $x$, essentially the routing probability ($\mu_l$) determines the class of $x$.

## 3.2 Max Route-based Confidence Measure

Based on the above description, given a sample image $x$, each decision tree in the forest produces a probability distribution $\mu(x|\Theta)$ over each *route*, or path from root-to-leaf, where

$\mu_l(x|\Theta)$ represents the probability that sample $x$ reaches leaf $l$. More specifically, $\mu_l(x|\Theta)$ is expressed as follows:

$$\mu_l(x|\Theta) = \prod_{n \in \mathcal{N}} d_n(x;\Theta)^{\mathbb{1}_{l \swarrow n}} \bar{d}_n(x;\Theta)^{\mathbb{1}_{l \searrow n}} \tag{3}$$

where $\bar{d}_n(x;\Theta) = 1 - d_n(x;\Theta)$, and $\mathbb{1}_{l \swarrow n}$ is 1 if leaf $l$ belongs to the left subtree of node $n$ and 0 otherwise. Similarly, $\mathbb{1}_{l \searrow n}$ is 1 if leaf $l$ belongs to the right subtree of node $n$ and 0 otherwise.

As shown in Eq. 3, the probability of a route is computed as the product of the probabilities w.r.t. all split nodes on that route. Therefore, given an image $x$, the probability distribution of the routes is expected to be very skewed and biased to one particular route. A given route will have an extremely small probability if $x$ does not fit the features represented by the split nodes on this route learned through the deep CNN, because the product of multiple small probabilities will diminish quickly. Typically just one route will stand out if all its split nodes match the features of $x$ well. We call this route the *max route*, because it has the maximum probability. It determines the class of $x$. The probability of the max route (or *max route probability*) can be used to measure how confident the classifier is about its classification decision of image $x$. The larger the max route probability is, the more confident the classifier is about the image. More specially, given an image $x$ and a decision tree $T$, the confidence is measured as:

$$\mathbb{C}_T(x;\Theta) = max\{\mu_l(x|\Theta)|l \in \mathcal{L}\} \tag{4}$$

where $max\{\mu_l(x|\Theta)|l \in \mathcal{L}\}$ denotes the max route probability of decision tree $T$ for the given image $x$. Since a forest is an ensemble of decision trees, the final confidence of image $x$ is measured as:

$$\mathbb{C}_{\mathcal{F}}(x) = \sum_{h=1}^{k} \mathbb{C}_{T_h}(x;\Theta_h) \tag{5}$$

**Discussion of the Effectiveness in Outlier Detection.** Intuitively this max route probability can be expected to be more effective than the maximum weighted sum of the classes of the CNN at separating outliers, as an outlier image will not have a good match with every split node on the max route. So its max route probability is limited by the product operation in the computation. In contrast, the maximum weighted sum in CNN tends to fall off much more slowly because the score is computed based on the linear combination of multiple features, such that a single matching feature can make the score high. This is also confirmed in our experimental studies.

### 3.3 Further Optimization: Regularization on Routing Decisions

To further improve the effectiveness of using the max route probability to detect outliers, we introduce a regularization to prevent the generation of routing decision whose probability distribution is close to *uniform*. We use a information theory-based approach. That is, we penalize the routing decision whose probability distribution has a large entropy, because a uniform distribution has a large entropy. This ensures that the max route probability of each routing decision always stands out, making it more effective at detecting outliers.

More specifically, given a decision tree $T$, $|\mathcal{L}|$ routing options exist for each sample $x \in \mathcal{X}$ reaching different prediction nodes. The entropy of the route probability distribution of sample $x$ is given by:

$$H(\mu(x|\Theta)) = -\sum_i \mu_i(x|\Theta)log(\mu_i(x|\Theta)), \tag{6}$$

where $\mu_i(x|\Theta)$ denotes the probability that sample $x$ reaching leaf $l_i$. However, the learning process does not converge during runtime if we directly apply this penalty function on the routing probability distribution due to numerical instability. We solve this problem by applying a softmax function on the probability distribution as a normalization. The revised entropy of the routing probability distribution is then given by:

$$H(\mu'(x|\Theta)) = -\sum_i \mu_i'(x|\Theta)log(\mu_i'(x|\Theta)), \qquad (7)$$

where $\mu_i'(x|\Theta) = Softmax(\mu_i(x|\Theta)) = \frac{exp(\mu_i(x|\Theta))}{\sum_j exp(\mu_j(x|\Theta))}$ is the softmax transformation of $\mu_i(x|\Theta)$.

To penalize the routing whose probability distribution has a large entropy, we add the entropy w.r.t. each training sample to the log-loss term. Given a sample $x \in \mathcal{X}$ and an output distribution $y \in \mathcal{Y}$, the log-loss term of one decision tree $T$ is represented as:

$$L(\Theta, \pi; x, y) = -log(\mathbb{P}_T[y|x, \Theta, \pi]) + \beta H(\mu'(x|\Theta)) \qquad (8)$$

where $\beta$ controls the strength of the penalty and $\mathbb{P}_T[y|x, \Theta, \pi]$ is defined in Eq. 2.

Therefore, the total log-loss for the random forest composed of $|\mathcal{T}|$ trees is defined as:

$$L(\Theta, \pi; \mathcal{T}) = \frac{1}{|\mathcal{T}|} \sum_{(x,y) \in \mathcal{T}} L(\Theta, \pi; x, y), \qquad (9)$$

Fig. 1(b) shows the max route probability distributions w.r.t. CIFAR-10 and CIFAR-100 tested on the CIFAR-10 model. By setting k as 5000, we get a max route probability cutoff threshold $ct = 0.99864$. As confirmed, 95% of CIFAR-100 images have their max route probabilities *smaller* than this large $ct$. Therefore, it achieves an outlier detection accuracy around 0.95, making it much more effective than using a threshold on CNN class weights.

## 4 ACCURACY IN CLASSIFICATION

Although skewness in the classification probability distribution (for example, the output of softmax in CNN) can benefit the detection of outlier images, it is known to hurt the generalization of image classifiers due to the high risk of overfitting (Pereyra et al., 2017). The situation seems worse for our deep neural decision forest-based approach, because the product operation in the routing probability computation makes the probability distribution of the routing more skewed compared to the output of softmax in CNN.

We overcome this in our decision forest-based approach as a result of the ensemble nature of the decision forest, allowing us to maintain good generalization performance. This is because, although one tree may overfit some aspects of the images, the ensemble provides a tool to make a composite prediction so that the overfitting in each individual tree is overcome.

More specifically, as an ensemble of decision trees $\mathcal{F} = \{T_1, \ldots, T_k\}$, a forest $\mathcal{F}$ delivers a prediction for a sample $x$ by averaging on the output of each tree, i.e.,

$$\mathbb{P}_{\mathcal{F}}[y|x] = \frac{1}{k} \sum_{h=1}^{k} \mathbb{P}_{T_h}[y|x] \qquad (10)$$

**Enhancement of Generalization Capability.** Next, we introduce a new architecture of the deep neural decision forest to further enhance its generalization capability. As shown in Fig. 2, although in (Kontschieder et al., 2015), different trees are connected to different sets of nodes of the *final* FC layer, the trees still share the other FC layers of the CNNs. This violates the principle of random decision forest where the trees in a forest should be as independent as possible so that that they can compensate each other. That is why decision forests tend to show excellent generalization capability. Therefore, we design a new

architecture (Fig. 3, Appendix A). It divides all FC layers into $k$ independent components, each of which is connected to one individual tree. This ensures independence amongst the trees and in turn improves the classification accuracy.

In summary, regularization and our new architecture together enable our deep neural decision forest-based approach to provide high accuracy at image classification while also assuring effective outlier detection, as we demonstrate in our experiments (Sec. 5).

In Appendix B, we show how to train the deep neural decision tree. The training requires estimating both the decision node parametrizations $\theta$ and the leaf predictions $\pi$. In (Kontschieder et al., 2015) the minimum empirical risk principle under log-loss is adopted for estimation by using a two-step optimization strategy, where $\theta$ and $\pi$ are updated alternatively to minimize the log-loss. In this work we use a new form of prediction nodes that makes the deep neural forest fully differentiable. This enables us to abandon the two-step optimization strategy and instead optimize the $\Theta$ and $\Pi$ parameters jointly in one step through back-propagation. This back-propagation based learning process is also described in Appendix B.

## 5 EXPERIMENTAL EVALUATION

### 5.1 OVERVIEW OF EXPERIMENTAL SETTING

**Datasets.** We empirically demonstrate the effectiveness of our proposed image outlier detection (IOD) strategy on several benchmark image datasets. Specifically, we train models on CIFAR-10 (Krizhevsky & Hinton, 2009) and MNIST (LeCun et al., 1998) datasets. Given a trained model on one data set, we consider examples from other datasets as outliers when testing the model. For example given a CIFAR-10 model, images in CIFAR-100, SVHN (Netzer et al., 2011), and MNIST datasets are outliers, because the images in these datasets have features distinct from those common to the CIFAR-10 images captured by the classification model.

**Methodology.** We evaluate: (1) *IOD-IOD-Weighted-Sum* described in Sec. 2. The weighted sum score in the final FC layer of the traditional CNN is utilized as the confidence measure; (2) *IOD-IOD-Max-Route-Shared-FC* method based on the original deep neural forest architecture proposed in (Kontschieder et al., 2015) in which each tree shares the FC layers as described in Sec. 3.1. Here, the max route is used as the confidence measure of IOD; (3) *IOD-Max-Route-Different-FC*: it uses our new deep neural forest architecture. Different trees are connected to isolated FC layers. Similarly the max route is used as the confidence measure; (4) *IOD-Max-Route-Penalty*: it uses our new deep neural forest architecture with a regularization term applied to the loss function (Eq. 8); (5) Deep SVDD (Ruff et al., 2018): the state-of-art fully deep once-class method described in Sec. 6; (6) AnoGAN (Schlegl et al., 2017): the state-of-art generative approach based on GAN. We compare against Deep SVDD and AnoGAN because they focus on image datasets.

**Experimental Setup.** We ran experiments on a GPU. All IOD models are implemented based on Pytorch (Paszke et al., 2017). For Deep SVDD and AnoGAN we reuse the codes provided by their authors. The code and models will be made public available via Github.

**Metrics.** We measure the accuracy of outlier detection and the classifation accuracy of our IOD-based approaches.

**Parameter Settings.** We set learning rates manually. The detailed settings of the parameters are provided in each subsection. All networks are trained using mini-batches of size 128. The momentum is set to 0.9 for all models. The weighted decays are set to 0.0001. We do not use data augmentation. We use simple global contrast normalization to pre-process all images. When testing CIFAR-10, CIFAR-100, and SVHN datasets on the model trained for MNIST, we change the image to gray scale and take central crops of the images. On the other hand, when testing MNIST on models trained for other datasets, we increase its color channel from 1 to 3 by copying the original gray image 3 times.

## 5.2 CIFAR-10

The CIFAR-10 dataset (Krizhevsky & Hinton, 2009) is composed of 10 classes of natural images with 50,000 training images and 10,000 testing images. Each image is an RGB image of size $32 \times 32$.

We use a deep NN composed of 10 convolutional layers and 2 FC layers with 1024 and 384 hidden units correspondingly. For the IOD-Weighted-Sum method, a 10-way linear layer is used for the final prediction. For the other three max route-based approaches, the output is connected to a deep neural forest containing 20 depth-3 trees. Specifically, for the IOD-Max-Route-Shared-FC method, the output of the first FC layer using 384 hidden units is connected to the second FC layer with $300 \ (= 20 \times (2^{(3+1)} - 1))$ hidden nodes. It is then connected to 20 depth-3 trees. For the IOD-Max-Route-Different-FC and the IOD-Max-Route-Penalty methods, the output of the convolutional layer is broadcast to 20 different sets of FC layers. Each set contains 2 FC layers using 1024 and 384 hidden units. Each connects to the final FC layer with $15 \ (= 2^{(3+1)} - 1)$ hidden nodes. This final FC layer is then connected to a tree. The learning rate is initialized as 0.1 and eventually decays to 0.001. For DEEP SVDD and AnoGAN we use the settings recommended by the authors.

The parameter k is set as 5000. As shown in Sec. 2 $k$ is used to establish an outlierness cut off threshold $ct$. For our own IOD approach, a testing image is an outlier if its maximum weighted sum score or max route probability is smaller than $ct$. For DEEP SVDD and AnoGAN an image is an outlier if its outlierness score is larger than the corresponding $ct$.

Table 1: CIFAR-10 Results.

| Methods | % Outliers Classified as Outliers | | | Training Accuracy | Testing Accuracy |
|---|---|---|---|---|---|
| | CIFAR-100 | SVHN | MNIST | | |
| IOD-Weighted-Sum | 69.61% | 82% | 24.6% | 99.830% | 88.970% |
| IOD-Max-Route-Shared-FC | 84.63% | 94.90% | 67.59% | 99.936% | 89.260% |
| IOD-Max-Route-Different-FC | 90.51% | 97.46% | 79.69% | 100% | 89.830% |
| IOD-Max-Route-Penalty | 94.69% | 97.44% | 94.94% | 99.982% | 88.030% |
| Deep SVDD (Ruff et al., 2018) | 20.16% | 14.8% | 50.12% | XX%[a] | XX% |
| AnoGAN (Schlegl et al., 2017) | 16.30% | 12.90% | 90.43% | XX% | XX% |

[a] Deep SVDD and AnoGAN do not have classification feature.

In comparison to the state-of-the-art (Table 1), our IOD-Max-Route-Penalty method significantly outperforms Deep SVDD and AnoGAN in all cases. Deep SVDD performs poorly, because as a one-class method it is not good at separating outliers from inliers that belong to multiple normal classes. AnoGAN only works well in detecting the MNIST images as outliers which are significantly different from CIFAR-10, while fails in all other cases when the outliers share more features with the inliers.

As for our own IOD-based methods, our max route-based methods significantly outperform IOD-Weighted-Sum that uses weighted-sum as confidence measure to detect outliers, without giving up our ability to correctly classify the CIFAR-10 images. The performance gain results from the deep neural forest architecture, in which the confidence w.r.t. each image is computed as the multiplication of the routing probabilities produced at the decision nodes on the route. Compared to the weighted sum approach, this multiplication of probabilities enlarges the confidence gap. Thus it is able to better detect outliers than the weighted sum approach. In addition, two of our max route-based methods (IOD-Max-Route-Shared-FC and IOD-Max-Route-Different-FC) achieve even better classification accuracy compared to the weighted sum-based method that uses the classical CNN architecture.

The IOD-Max-Route-Penalty method outperforms the other two IOD-Max-Route based methods in detecting outliers in almost all cases, especially in detecting MNIST outliers. This is because by introducing regularization to penalize routing decisions that have large entropy, IOD-Max-Route-Penalty avoids uniform route probability distributions and thus leads to larger maximum route probabilities for inliers. At the same time, its classification accuracy decreases a little bit, because the regularization might introduce overfitting (Szegedy et al., 2016). However, it is effectively alleviated because of the ensemble of the decision trees. Therefore, the drop on the classification accuracy is almost negligible.

Moreover, IOD-Max-Route-Different-FC consistently outperforms IOD-Max-Route-Shared-FC in both outlier detection and classification. This demonstrates the effectiveness of our new proposed deep neural forest architecture as compared to the original version. This new architecture has a large capacity in detecting outliers and classifying images because of the fully isolated decision trees in the forest.

### 5.3 MNIST

Due to space limitation, please refer to Appendix C for the results on MNIST.

## 6 RELATED WORK

**Classical Outlier Detection.** Outlier detection has been extensively studied in the literature (Breunig et al., 2000; Knorr & Ng, 1998; Ramaswamy et al., 2000; Bay & Schwabacher, 2003). These outlier detection techniques in general share one common principle. Namely, an object is an outlier if its outlierness score is above a certain cutoff threshold. This principle is also leveraged in our context. However, unlike these works (Breunig et al., 2000; Knorr & Ng, 1998; Ramaswamy et al., 2000; Bay & Schwabacher, 2003), the outlierness score is not measured in the original feature space. This avoids the similarity search problem in the high dimensional image data.

**One-Class Classification.** One-class Support Vector Machine (OC-SVM) and Support Vector Data Description (SVDD) methods (Schölkopf et al., 2001; Manevitz & Yousef, 2002; Tax & Duin, 2004) use only normal training data to separate outliers from normal data based on kernel-based transformations. In particular, OC-SVM uses a hyperplane (Schölkopf et al., 2001; Manevitz & Yousef, 2002) to separate the target objects from the origin with maximal margin. It is based on the assumption that the origin of a kernel-based transformed representation belongs to the outlier class. SVDD is a different kernel SVM method in which data is enclosed in a hypersphere of radius $R$ in the transformed feature space. The squared radius is minimized together with penalties for margin violation. However, these approaches are shown to be extremely sensitive to specific choices of the representation, kernel, and the hyper-parameters. The difference in performance can be dramatic based on these choices (Manevitz & Yousef, 2007). Therefore, these methods are not robust. In addition, they assumed that all normal data belongs to one single class. This does not fit our scenario in which the applications generate rich classes of images.

**Shallow Image Outlier Solutions.** In Shamir (2013), the authors proposed an unsupervised image outlier method. It first extracts a set of image features from the raw pixels and several combinations of image transforms. Then, the mean and the variance values of each feature are computed over all images in the dataset. An image is considered as an outlier if the values of its features significantly deviate from their mean values. This approach performs poorly even on the simplistic MNIST due to its limited feature extraction ability. In Ju et al. (2015), the authors proposed a probabilistic PCA model focusing on the extraction of the features that result in accurate image reconstruction. During this process, the outliers are identified and removed to improve the performance of the PCA model. Therefore, the target of this work is on extracting the most typical features instead of outlier detection.

**Deep Outlier Detection Solutions.**

*Deep One-class methods.* Research efforts have been put on enhancing one-class classification with the representation learning ability of deep neural network. In Perera & Patel (2018); Erfani et al. (2016), "mix" methods have been proposed that detect outliers by first learning a representation using deep learning and then feed that into a classical shallow outlier detection method. Recently, "fully deep" methods (Ruff et al., 2018; Nguyen & Vien, 2018) were proposed that produce representations and the boundary separating outliers from inliers (hyperplane in OC-SVM and hypersphere in SVDD) within one learning process. These methods outperform the mix methods according to Ruff et al. (2018). Unfortunately, these methods still suffer from the fundamental problem of one-class classification, namely they only find a single boundary between outliers and inliers. However, real datasets tend to have multiple classes of normal images. For multi-class datasets, it is difficult to find a

separator that encompasses all normal classes and none of the outliers. As confirmed in our experiments, our IOD method significantly outperforms the one-class approaches when handling complex image datasets such as CIFAR-10.

*Deep Autoencoder-based Methods.* Deep autoencoder has been broadly used to detect outliers. It is based on the observation that these networks are able to extract the common factors of variation from normal samples and reconstruct them accurately, while anomalous samples do not contain these common factors of variation and thus cannot be reconstructed accurately. The deep autoencoders can be used to detect outliers in a mixed approach (Erfani et al., 2016), namely by plugging in the learned embeddings into classical outlier detection methods. It can also be directly used to detect outliers by employing the reconstruction error as the outlier score (Zhou & Paffenroth, 2017; Chen et al., 2017). However, the objective of autoencoders is dimension reduction. It does not target on outlier detection directly. Therefore, the learned low dimensional representation is not necessarily effective in distinguishing outliers from inliers. Furthermore, choosing the right degree of compression is also difficult when applying autoencoders for outlier detection.

*GAN-based Methods.* The GAN-based method (Schlegl et al., 2017) (AnoGAN) first trains a GAN to generate samples according to the training data. A test object is an outlier if it cannot find a close point in the generator's latent space. However, similar to autoencoders, generative approaches have difficulty in regularizing the generator for compactness.

*Deep Statistical Methods.* The deep statistical methods detect outliers by directly modeling the data distribution with the help of deep structures such as autoencoder. In particular, in Zong et al. (2018) the Deep Autoencoding Gaussian Mixture Model (DAGMM) method uses a deep autoencoder to generate a low-dimensional representation and reconstruction error which are then used to build a Gaussian Mixture Model (GMM). The parameters of the deep autoencoder and the mixture model are jointly optimized in an end-to-end fashion. Similarly, in Zhai et al. (2016) the deep structured energy based models (DSEBMs) detects outliers by connecting the autoencoder and the energy based model (EBM). However, the statistical methods detect outliers purely based the density of the objects. The model might assign high density to the outliers if there are many proximate outliers, hence resulting in false negative.

**Open Set Deep Network.** In Bendale & Boult (2016) a method was introduced to adapt Convolutional Neural Networks to discover a new class by adding one additional class at the softmax layer. It leverages the weighted sum score to determine whether one image belongs to the new class. However, its techniques are tightly coupled with one specific network architecture, namely AlexNet (Krizhevsky et al., 2012a). Our *IOD* framework instead can accommodate any type of image classifiers.

**Learning from Noisy Labeled Image Data.** In Xiao et al. (2015), the authors studied the problem of training CNNs that are robust to a large number of noisy labels. The idea is to extend CNNs with a probabilistic model, which infers the true labels from the noisy labels and then uses them as clean labels in the training of the network. This assumes that the noisy labels or outliers are already known. Therefore, this is not only totally different from our outlier detection problem, but also not practical in most applications where outliers are unknown and unexpected phenomenon.

## 7 CONCLUSION

In this work we propose a novel approach that effectively detects outliers from image data. The key novelties include a general image outlier detection framework and effective outlierness measure that leverages the deep neural decision forest. Optimizations such as new architecture that connects deep neural network and decision tree and regularization to penalize the large entropy routing decisions are also proposed to further enhance the outlier detection capacity of IOD. In the future we plan to investigate how to make our approach work in multi-label classification setting.

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

## A  New Decision Forest Architecture

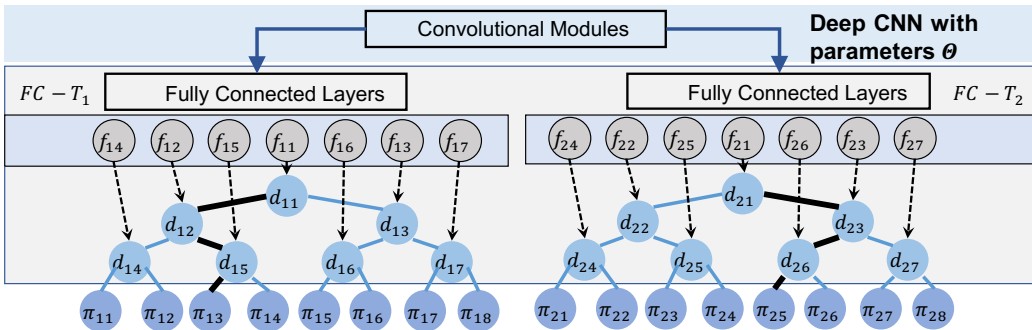

Figure 3: New Decision Forest Architecture.

## B  The learning of the Deep Neural Decision Forest

Learning a deep neural decision tree requires estimating both the decision node parametrizations $\theta$ and the leaf predictions $\pi$. In Kontschieder et al. (2015) the minimum empirical risk principle under log-loss is adopted for their estimation by using a two-step optimization strategy, where $\theta$ and $\pi$ are updated alternatively to minimize the log-loss.

In this section we first introduce a new form of prediction nodes that makes the deep neural forest fully differentiable. This enables us to abandon the two-step optimization strategy and instead optimize the $\Theta$ and $\Pi$ parameters jointly in one step through back-propagation. Next, we show the back-propagation based learning process.

### B.1  New Prediction nodes

The prediction node used in Kontschieder et al. (2015) holds a probability distribution $\pi_l$ over $\mathcal{Y}$ and are optimized alternately with the decision nodes $d$. Specifically, the optimal predictions for all prediction nodes can be obtained by minimizing a convex objective given fixed decision nodes, while the parameters in the decision nodes are trained through the back-propagation process in each training epoch. Obviously, this two-step strategy is not efficient. Here, we use new forms of prediction nodes that makes the decision forest fully differentiable and therefore can be optimized jointly with the decision nodes as shown in Sec. B.2.

Each prediction node is parametrized using a $k$-dimensional parametric probability distribution $w_l$, denoted as:

$$\pi_l = softmax(w_l) = \frac{e^{w_{l_i}}}{\sum_{j=1}^{k} e^{w_{l_j}}} \tag{11}$$

where $k$ is the number of classes. The softmax function takes a vector of real-valued scores and converts it to a vector of values between 0 and 1 that sum to one. In Sec. B.2, we will show that using the new prediction nodes, the final loss function is still convex.

### B.2  Learning Forest by Back-Propagation

Learning the random forest model requires us to find a set of parameters $\Theta$ and $\pi$ that can minimize the total log loss defined in Eq. 9. To minimize Eq. 9, in fact we only need to independently minimize the penalized loss (Eq. 8) of each individual tree. Next, we show that the loss function is fully differentiable. Therefore, we are able to employ a Stochastic Gradient Descent (SGD) approach to minimize the loss w.r.t. $\Theta$ and $\pi$, following the common practice of back-propagation in deep neural networks.

**Learning Decision Nodes by Back-Propagation.** Given a decision tree, the gradient of the loss $L$ with respect to $\Theta$ can be decomposed by the chain rule as follows:

$$\frac{\partial L}{\partial \Theta}(\Theta, \pi; x, y) = \sum_{n \in \mathcal{N}} \frac{\partial L(\Theta, \pi; x, y)}{\partial f_n(x; \Theta)} \frac{\partial f_n(x; \Theta)}{\partial \Theta} \tag{12}$$

Here, the derivative of the second part $\frac{\partial f_n(x;\Theta)}{\partial \Theta}$ is identical to the back-propagation process of traditional CNN modules and thus is omit here. Now let's show how to compute the first part:

$$\begin{aligned}
\frac{\partial L(\Theta, \pi; x, y)}{\partial f_n(x; \Theta)} &= \frac{\partial(-log(\mathbb{P}_T[y|x, \Theta, \pi]) + \beta H(\mu'(x|\Theta)))}{\partial f_n(x; \Theta)} \\
&= \frac{\partial(-log(\mathbb{P}_T[y|x, \Theta, \pi])}{\partial f_n(x; \Theta)} + \beta \frac{\partial(H(\mu'(x|\Theta)))}{\partial f_n(x; \Theta))}
\end{aligned} \tag{13}$$

where

$$\frac{\partial(-log(\mathbb{P}_T[y|x, \Theta, \pi])}{\partial f_n(x; \Theta)} = -\sum_{l \in \mathcal{L}} \frac{\pi_{ly}}{\mathbb{P}_T[y|x, \Theta, \pi]} \frac{\partial \mu_l(x|\Theta)}{\partial f_n(x; \Theta)} \tag{14}$$

and

$$\begin{aligned}
\sum_{l \in \mathcal{L}} \frac{\partial \mu_l(x|\Theta)}{\partial f_n(x; \Theta)} &= -\sum_{l \in \mathcal{L}} \mu_l(x|\Theta) \frac{\partial log(\mu_l(x|\Theta))}{\partial f_n(x; \Theta)} \\
&= -\sum_{l \in \mathcal{L}} \mu_l(x|\Theta)(\mathbb{1}_{l \swarrow n} \bar{d}_n(x; \Theta) - \mathbb{1}_{l \searrow n} d_n(x; \Theta)) \\
&= -\sum_{l \in \mathcal{L}_{n_l}} \mu_l(x|\Theta) \bar{d}_n(x; \Theta) + \sum_{l \in \mathcal{L}_{n_r}} \mu_l(x|\Theta) d_n(x; \Theta)
\end{aligned} \tag{15}$$

By using the chain rule

$$\frac{\partial(H(\mu'(x|\Theta)))}{\partial f_n(x; \Theta))} = \sum_{l \in \mathcal{L}} \frac{\partial(H(\mu'(x|\Theta)))}{\partial(\mu_l'(x|\Theta))} \frac{\partial(\mu_l'(x|\Theta))}{\partial(\mu_l(x|\Theta))} \frac{\partial(\mu_l(x|\Theta))}{\partial f_n(x; \Theta))} \tag{16}$$

where

$$\begin{aligned}
\sum_{l \in \mathcal{L}} \frac{\partial(H(\mu'(x|\Theta)))}{\partial(\mu_l'(x|\Theta))} &= \sum_{l \in \mathcal{L}} \frac{\partial(\mu_l'(x|\Theta) log(\mu_l'(x|\Theta)))}{\partial(\mu_l'(x|\Theta))} \\
&= \sum_{l \in \mathcal{L}} (1 + log(\mu_l'(x|\Theta)))
\end{aligned} \tag{17}$$

and

$$\sum_{l \in \mathcal{L}} \frac{\partial(\mu_l'(x|\Theta))}{\partial(\mu_l(x|\Theta))} = \sum_{l \in \mathcal{L}} (s_l(1 - s_l) + \sum_{k \neq l} s_l s_k)$$

where

$$s_l = \frac{e^{\mu_l(x|\Theta)}}{\sum_{k \in \mathcal{L}} e^{\mu_k(x|\Theta)}}$$

**Learning Prediction Nodes by Back-Propagation.** Given a decision tree, the gradient of the Loss $L$ w.r.t. the weights $w$ of the prediction nodes (defined in Eq. 11) can be decomposed by the chain rule as follows:

$$\frac{\partial L}{\partial w}(\Theta, \pi; x, y) = \sum_{l \in \mathcal{L}} \frac{\partial L(\Theta, \pi; x, y)}{\partial \pi_l} \frac{\partial \pi_l}{\partial w_l} \tag{18}$$

where

$$\frac{\partial L(\Theta, \pi; x, y)}{\partial \pi_{ly}} = -\frac{\mu_l(x|\Theta)}{\mathbb{P}_T[y|x, \Theta, \pi])} \quad (19)$$

and

$$\frac{\partial \pi_{ly}}{\partial w_{li}} = \begin{cases} \pi_{ly}(1 - \pi_{ly}) & y = i \\ -\pi_{ly}\pi_{li} & y \neq i \end{cases} \quad (20)$$

Therefore,

$$\begin{aligned} \frac{\partial L(\Theta, \pi; x, y)}{\partial w_{li}} &= \sum_{y \in \mathcal{Y}} \frac{\partial L(\Theta, \pi; x, y)}{\partial \pi_{ly}} \frac{\partial \pi_{ly}}{\partial w_{li}} \\ &= \frac{\partial L(\Theta, \pi; x, i)}{\partial \pi_{li}} \frac{\partial \pi_{li}}{\partial w_{li}} + \sum_{y \neq i} \frac{\partial L(\Theta, \pi; x, y)}{\partial \pi_{ly}} \frac{\partial \pi_{ly}}{\partial w_{li}} \\ &= -\frac{\mu_l(x|\Theta)(\pi_{li}(1 - \pi_{li}))}{\mathbb{P}_T[i|x, \Theta, \pi])} + \sum_{y \neq i} \frac{\mu_l(x|\Theta)\pi_{ly}\pi_{li}}{\mathbb{P}_T[y|x, \Theta, \pi])} \end{aligned} \quad (21)$$

## C    Experiment Results on MNIST

MNIST (LeCun et al., 1998) consists of $28 \times 28$ pixel grayscale images of handwritten digits from 0 to 9. There are 60,000 training images and 10,000 testing images in the dataset. In this case we use a neural network composed of 3 convolutional layers and one FC layer with 625 hidden units. Similar to the CIFAR-10 model, the weighted-sum method uses a 10-way softmax layer for the final prediction, while the max route-based methods connect the neural network to a decision forest containing 5 depth-8 trees. The initial learning rate is set as 0.0001 for the weighted-sum method and 0.001 for other max route-based methods. A multi-step decay is then applied in 50 epochs. The number of outliers in MNIST is set to 200 ($k = 200$) to get the confidence cutoff threshold (ct) (Sec. 2).

Table 2: MNIST Results.

| Methods | % outliers classified as outliers | | | Training Accuracy | Testing Accuracy |
|---|---|---|---|---|---|
| | CIFAR-10 | CIFAR-100 | SVHN | | |
| IOD-Weighted-Sum | 81.53% | 83.25% | 83.99% | 99.725% | 99.240% |
| IOD-Max-Route-Shared-FC | 90.83% | 93.11% | 94.09% | 99.877% | 99.440% |
| IOD-Max-Route-Different-FC | 91.35% | 94.04% | 93.03% | 99.743% | 99.49% |
| IOD-Max-Route-Penalty | 94.01% | 96.12% | 94.85% | 99.780% | 99.400% |
| Deep SVDD (Ruff et al., 2018) | 94.73% | 94.565% | 94.49% | XX% | XX% |
| AnoGAN (Schlegl et al., 2017) | 96.50% | 95.10% | 88.67% | XX% | XX% |

As shown in Table 2, similar to the CIFAR-10 experiments, our max route-based IOD methods significantly outperforms the weighted-sum based IOD method in both outlier detection and classification due to the same reason discussed in the CIFAR-10 experiments.

As for our three max route-based IOD approaches, the IOD-Max-Route-Penalty approach again outperforms the other two approaches in outlier detection in all cases, while only negligible classification accuracy is sacrificed. Moreover, IOD-Max-Route-Different-FC consistently outperforms IOD-Max-Route-Shared-FC in both outlier detection and classification because of the new deep neural forest architecture.

Deep SVDD and AnoGAN perform well on this simplistic MNIST dataset, because the testing datasets (CIFAR-10, CIFAR-100, and SVHN) which are treated as outliers are significantly different from MNIST as normal data.

## D   The Evaluation Results on Dropout Method

In Gal & Ghahramani (2016), a new theoretical framework was proposed that casts dropout used in training deep neural networks as approximate Bayesian inference in deep Gaussian processes. A direct result of this theory gives us a tool to model uncertainty with dropout NNs. As discussed in Gal & Ghahramani (2016), dropout is applied during inference. Given an image, we can get a range of softmax input values for each class by measuring 100 stochastic forward passes of the softmax input. If the range of the predicted class intersects that of other classes, then even though the softmax output is arbitrarily high (as far as 1 if the mean is far from the means of the other classes), the uncertainty of the softmax output can be as large as the entire space. In other words, the size of intersection signifies the model's uncertainty in its softmax output value – i.e., in the prediction. The larger the intersection is, the more uncertain the model is in its prediction.

Therefore, this uncertainty could be used as a score to measure the outlierness of the image. Therefore, we evaluates this dropout method as an additional baseline method.

**Evaluation on CIFAR-10 Data.**   We use the identical network architecture as suggested in the author's github repository (`https://github.com/yaringal/DropoutUncertaintyCaffeModels/tree/master/cifar10_uncertainty`). When the method is applied on the CIFAR-10 training data, for each CIFAR-10 image, we forward it in the model 100 times and record the softmax input values for each class. The "outlierness" (uncertainty) of the image is defined as the intersection between predict class and all other classes. We use the 5000-th largest value in the training as the cutoff threshold. Specifically, if an image has an uncertainty larger than the threshold, it is considered to be as an outlier. Then we forward each CIFAR-100 image 100 times in the model and record the outlier score for each CIFAR-100 images. Unfortunately, the accuracy of this outlier detection scheme is only 53%. That is, this is even worse than our maximum weighted sum baseline which has an accuracy of 70%.

**Evaluation on MNIST Data.** We also applied the Dropout method on MNIST training images. Again we use the network architecture suggested by the authors in their repository. We forward each MNIST image 100 times in the model and compute the outlierness score. However, when we use the 200-th largest outlierness score in MNIST training as the outlierness cutoff threshold (the same parameter setting to our MNIST experiment in Appendix C), the accuracy in detecting CIFAR-10 images (also forward 100 times in the model) as outliers is lower than 10%. When we increase the parameter from 200 to 5000, its accuracy in detecting outliers increases to 48.13%, which is still much much lower than our proposed method (above 90%), although clearly in this case the parameter setting biases to Dropout method.

## E   The Experiment Results of Varying Parameter $k$

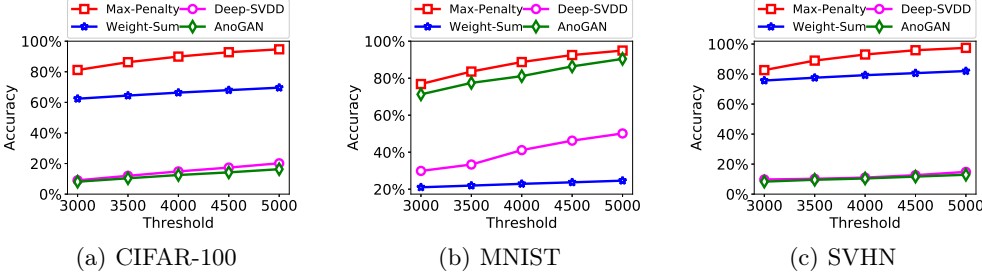

(a) CIFAR-100          (b) MNIST          (c) SVHN

Figure 4: Outlier Detection Accuracy on CIFAR-10 Model: Varying k

We evaluate how the input parameter k influences the accuracy of outlier detection. As discussed in Sec. 2, in our IOD-based method, k is used to establish a confidence cut off threshold that corresponds to the kth smallest maximum route probability or maximum

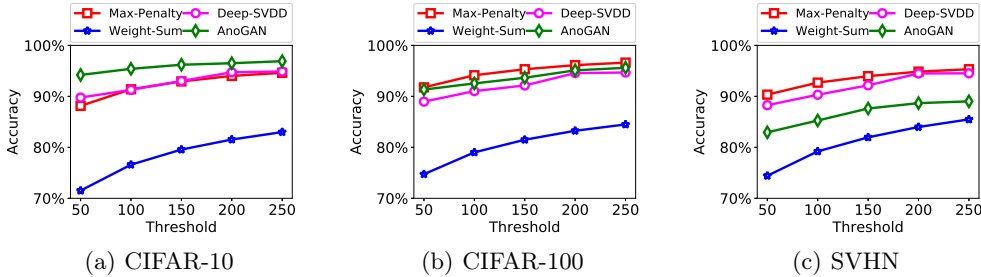

Figure 5: Outlier Detection Accuracy on MNIST Model: Varying k

weighted sum among the training objects. While in Deep SVDD and AnoGAN, it corresponds to the kth largest outlierness score among the normal objects. In our IOD-based method, given a testing image, if its maximum weighted sum or maximum route probability is smaller than the cutoff threshold, it is considered as an outlier. In Deep SVDD and AnoGAN, a testing image is considered as an outlier if its outierness score produced by Deep SVDD or AnoGAN is larger than the corresponding cutoff threshold.

Figure 4 shows the results on the CIFAR-10 model. Our IOD-based IOD-Max-Route-Penalty (Max-Penalty) method significantly outperforms the state-of-art SVDD and AnoGAN methods in all cases. When $k$ decreases, all methods have a lower accuracy of outlier detection. This is expected, because a smaller k produces a smaller cut off threshold in our IOD-based method and a larger cut off in Deep SVDD and AnoGAN, both leading to the detection of smaller number of outliers.

Similarly, as shown in Figure 5, our IOD-Max-Route-Penalty (Max-Penalty) consistently outperforms all other methods in almost all cases when detecting outiers out of the simpler MNIST model. The only exception is that AnoGAN is slightly better than Max-Penalty when detecting CIFAR-10 outliers out of MNIST model.

## F    THE EVALUATION ON ISOLATION FOREST

Isolation Forest Liu et al. (2008) is a widely used unsupervised outlier detection method. It builds an ensemble of isolation trees for a given data set. Then the average path lengths on the isolation trees is used as the outlierness measurement. The shorter the average path length is, the more likely the instance is to be an outlier.

**Evaluation on CIFAR-10 Data.** First, we used a CNN to extract features from the raw image data and then build an Isolation Forest on these extracted features to detect outliers. As one input parameter of Isolation Forest, the number of outliers in CIFAR-10 is set as 5000 – identical to the parameter $k$ used in our IOD-based methods.More specifically, similar to our IOD-based approach, we use the 5,000th smallest average path length in CIFAR-10 as the cutoff threshold. If the average path length of a testing image is smaller than the cutoff threshold, it is considered an outlier. When the model is trained on CIFAR-10, we expect all images in CIFAR-100, MNIST and SVHN to be detected as outliers. However, in fact the outlier detection accuracy is poor – less than 2% in all cases, although we have carefully tuned the size of the network from producing 2042 dimensional feature vector to 512 dimensional feature vector and tuned other parameters in Isolation Forest such as the number of the trees and max_sample.

We also tried applying dimensionality reduction techniques to reduce the dimension of the extracted feature vectors and then applying Isolation Forests on the lower dimension space. The results were slightly better, although the detection rate for outliers is still lower than 2%.

Finally, we directly applied Isolation Forests on the raw image. The outlier detection accuracy with respect to CIFAR-100, SVHN, and MNIST was 9.73%, 13.14% and 8.3% respectively.

Although these results are much better than running against the extracted features from the CNN, they are still much worse than our maximum weighted sum baseline. Based on our evaluation, reducing the dimension of the raw features does not improve the outlier detection accuracy in this case.

**Evaluation on MNIST Data.** We also tested Isolation Forest on MNIST data. Identical to the evaluation of our IOD-based methods, the number of outlier is set as 200. Again, building the Isolation Forest on raw image data achieves the highest outlier detection accuracy, namely 66.2%, 59.78% and 58.47% with respect to CIFAR-10, CIFAR-100 and SVHN respectively. However, the results are still much worse than any method we have evaluated in our experimental section.

