# OpenReview forum: "Outlier Detection from Image Data"
_ICLR.cc/2019/Conference_

### Official Review · AnonReviewer2 · 2018-10-28
**Interesting promising solution to outlier detection; application of proposed scheme to general outlier detection seems limited**

**Rating:** 5
**Confidence:** 4

**Review:**

Pros
----

[Originality/Clarity]
The manuscript presents a novel technique for outlier detection in a supervised learning setting where something is considered an outlier if it is not a member of any of the "known" classes in the supervised learning problem at hand. The proposed solution builds upon an existing technique (deep neural forests). The authors clearly explain the enhancements proposed and the manuscript is quite easy to follow.

[Clarity/Significance]
The enhancements proposed are empirically evaluated in a manner that clearly shows the impact of the proposed schemes over the existing technique. For the data sets considered, the proposed schemes have demonstrated significant improvements for this scoped version of outlier detection.

[Significance]
The proposed scheme for improving the performance of the ensemble of the neural decision trees could be of independent interest in the supervised learning setting.

Limitations
-----------

[Significance]
Based on my familiarity with the traditional literature on outlier detection in an unsupervised setting, it would be helpful for me to have some motivation for this problem of outlier detection in a supervised setting. For example, the authors mention that this outlier detection problem might allow us to identify images which are incorrectly labelled as one of the "known" classes even though the image is not a true member of any of the known classes, and might subsequently require (manual) inspection. However, if this technique would actually be used in such a scenario, the parameters of the empirical evaluation, such as a threshold for outliers that considers 5000 images as outliers, seem unreasonable. Usually number of outliers (intended for manual inspection) are fairly low. Empirical evaluations with a smaller number of outliers is more meaningful and representative of a real application in my opinion.

[Significance]
Another somewhat related question I have is the applicability of this proposed outlier detection scheme in the unsupervised scheme where there are no labels and no classification task in the first place. Is the proposed scheme narrowly scoped to the supervised setting?

[Comments on empirical evaluations]
- While the proposed schemes of novel inlier-ness score (weighted sum vs. max route), novel regularization scheme and ensemble of less correlated neural decision trees are extremely interesting and do show great improvements over the considered existing schemes, it is not clear to me why the use of something like Isolation Forest (or other more traditional unsupervised outlier detection schemes such as nearest/farthest neighbour based) on the learned representations just before the softmax is not sufficient. This way, the classification performance of the network remains the same and the outlier detection is performed on the learned features (since the learned features are assumed to be a better representation of the images than the raw image features). The current results do not completely convince me that the proposed involved scheme is absolutely necessary for the considered task of outlier detection in a supervised setting.
- [minor] Along these lines, considering existing simple baselines such as auto-encoder based outlier detection should be considered to demonstrate the true utility of the proposed scheme. Reconstruction error is a fairly useful notion of outlier-ness. I acknowledge that I have considered the authors' argument that auto-encoders were formulated for dimensionality reduction.

[Minor questions]
- In Equation 10, it is not clear to me why (x,y) \in \mathcal{T}. I thought \mathcal{T} is the set of trees and (x,y) was the sample-label pair.
- It would be good understand if this proposed scheme is limited to the multiclass classification problem or is it also applicable to the multilabel classification problem (where each sample can have multiple labels).

---

> ### Author Response · Authors · 2018-11-26
> **Response to Reviewer 2 (Part 1):  comments on significance -- new evaluation with varying outlier threshold and the applicability of the proposed method**
>
>
> [COMMNENT FROM REVIEWER]: [Significance] A threshold that considers 5000 images as outliers seem unreasonable. Usually number of outliers intended for manual inspection is low.
>
> RESPONSE:
> With respect to the motivation, the key issue, as noted in our response to reviewer 3 above ("The extensibility of the proposed method"), is that an image classifier trained on a particular class of images may be exposed to images at inference time that contain objects that are from none of the classes the classifier was trained on. In such cases, the classifier will happily produce an output, and using simple confidence-based methods for rejecting such objects will often result in labeling new previously unseen objects as an existing class, sometimes with surprisingly high probability.  The main goal of our method is to reject such spurious detections.
>
> The threshold setting of 5000 as an input parameter is chosen for establishing an outlierness cutoff threshold for detecting outliers. It is used during the training phase. In other words, the users do not need to evaluate these 5000 images in the training set. We have updated Section 2 of our draft to avoid this confusion.
>
> As an input parameter, it can be set to either a larger or a smaller value per the need of the application. For example, if the user believes that the percentage of outliers in her application is large, then it should be set as a relatively large value. Otherwise it could be set to a smaller number. This is something an application domain expert would have to explore, given  this will vary based on the targeted application and its scope.
>
> However, to respond to your thoughts concerning the choice of an outlier threshold, we add charts into Appendix E to illustrate how the outlier detection accuracy changes as this input parameter varies. As discussed in Appendix E, our proposed method significantly outperforms the state-of-art in almost all cases, and does well across a range of thresholds.
>
>
> [COMMENT FROM REVIEWER]: [Significance] Is the proposed scheme narrowly scoped to the supervised setting?
>
> RESPONSE: Our approach does not rely on the labeled outliers to train an outlier classifier, although it needs labeled inliers to produce an outlierness score for each image and establish an outlierness cutoff threshold to detect outliers. As also noted in our response to reviewer 3 (The extensibility of the proposed method), we believe that our approach is broadly applicable in a rich variety of real world applications for two reasons:
>
> (1) It resolves a significant limitation of  traditional image classifiers. Given one testing image, an existing CNN image classifier will assign this image to one of the classes observed in the training set, even if it does not belong to any known class in the training data set. For example, given a cat image, if we test it on a CNN model trained using MNIST, this cat image will be erroneously assigned to one of the digit classes. In the real applications, it is common for images supplied at inference time to not belong to any class known in the training data  -- for example, consider an autonomous vehicle trained mostly on urban imagery taken to the desert, where it sees sand, cacti, and tumbleweed for the first time. Our approach thus enhances any of the existing CNN-based classifiers with this powerful ``rejection'' ability. That is, it no longer blindly assigns a testing image to one of the known classes. Instead, an image will be rejected as being an outlier if it does not ``sufficiently''  belong to any of the existing classes.
>
> (2) Real applications tend to have a sufficient amount of normal data, and thus are able to more easily provide us with a large amount of labeled normal data for training the classification model, while they lack access to labeled outliers due to the rarity of outliers. Thus, an approach, such as ours, that uses only labeled inliers, and does NOT rely on the availability of outlier labels is a preferred situation in practice.

---

> ### Author Response · Authors · 2018-11-26
> **Response to Reviewer 2 (Part 2): comments on empirical evaluation -- comparing against Isolation Forest**
>
>
> [COMMENT ON EMPIRICAL EVAL.] Why the use of something like Isolation Forest (IF) on the learned representations is not sufficient?
>
> RESPONSE: To address this comment, we tested the Isolation Forest-based  method as you suggest above on its ability of detecting image outliers. The Isolation Forest method first builds an ensemble of isolation trees for a given data set. Then the average path lengths on the isolation trees is used as the outlierness measurement. The shorter the average path length is, the more likely the instance is to be an outlier.
>
> First, per the suggestion of the reviewer, we used a CNN to extract features from the raw image data and then built an Isolation Forest on these extracted features to detect outliers. As one input parameter of Isolation Forest, the number of outliers in CIFAR-10 is set as 5000 -- identical to the parameter k used in our IOD-based methods. More specifically, similar to our IOD-based approach, we use the 5,000th smallest average path length in CIFAR-10 as the cutoff threshold. If the average path length of a testing image is smaller than the cutoff threshold, it is considered  an outlier. When the model is trained on CIFAR-10, we expect all images in CIFAR-100, MNIST and SVHN to be detected as outliers. However, in fact, the outlier detection accuracy is poor -- less than 2% in all cases, although we have carefully tuned the size of the network from producing 2042 dimensional feature vector to 512 dimensional feature vector and tuned other parameters in Isolation Forest such as the number of the trees and max_sample.
>
> We also tried applying dimensionality reduction techniques to reduce the dimension of the extracted feature vectors and then applying  Isolation Forests on the lower dimension space. The results were slightly better, although the detection rate for outliers is still lower than 2%.
>
> Finally, we directly applied Isolation Forests on the raw image. The outlier detection accuracy with respect to CIFAR-100, SVHN, and MNIST was 9.73%, 13.14% and 8.3% respectively. Although these results are much better than running against the extracted features from the CNN, they are still much worse than our maximum weighted sum baseline. Based on our evaluation, reducing the dimension of the raw features does not improve the outlier detection accuracy in this case.
>
> We also tested Isolation Forests on the simpler MNIST data. Identical to the evaluation of our IOD-based methods, the number of outliers is set as 200. Again, building the Isolation Forest on raw image data achieves the highest outlier detection accuracy, namely 66.2%, 59.78% and 58.47% with respect to CIFAR-10, CIFAR-100 and SVHN respectively. However, the results are still much worse than any method we have evaluated in Appendix C.

---

> ### Author Response · Authors · 2018-11-26
> **Response to Reviewer 3 (Part 3): other minor comments**
>
>
> [MINOR COMMENTS ]: auto-encoder based outlier detection.
>
> RESPONSE: As also introduced in our related work, auto-encoder techniques  have indeed been used for outlier detection. However, based on the evaluation and the discussion in [1], which is the state-of-the-art in image outlier detection, these methods do not perform better than the GAN-based methods in detecting outliers -- another popular type of generative model. Therefore, in this work we have chosen to compare against AnoGAN [2] as the representative method among this class of  generative models.
>
> [1] Ruff, Lukas, et al. ``Deep one-class classification.'' ICML 2018
>
> [2] Unsupervised Anomaly Detection with Generative Adversarial Networks to Guide Marker Discovery. IPMI 2017
>
> [MINOR COMMENTS]: In Equation 9 (previous Equation 10), it is not clear to me why (x,y) $\in$ $\mathcal{T}$.
>
> RESPONSE: Equation 9 demonstrates the total log loss of the forest, while the (x,y) demonstrates the training samples used to train the forest T during the training process. It has to be used when computing the total log loss.
>
> [MINOR COMMENTS]: if the proposed approach is also applicable to the multilabel classification problem (where each sample can have multiple labels).}
>
> RESPONSE: We thank the reviewer for this great question. While we have not targeted this more general scenario of multiple labels per instance, we describe below our reasoning for why we consider our IOD framework to also be applicable to this more general case.
>
> Theoretically, our image outlier detection (IOD) framework would still be applicable in multi-label classification scenarios. In principle, our proposed IOD framework decides on whether a testing image is an outlier or not based on the confidence of the classifier for this testing image. If the classifier has a small confidence about a given image, then IOD rejects making an assignment to that class. This confidence is measured based the probability that the classifier ``believes'' the testing image belongs to a particular class. In multi-label scenarios, each testing image is also assigned a probability with respect to each class typically by a sigmoid layer as opposed to the softmax layer used in a single-label classification. Therefore, IOD could still work.
>
> A simple solution would be to continue to establish a probability cutoff threshold based on the object that has the xth smallest probability in the training set, and then classify a testing image as an outlier if its largest probability produced by sigmoid is smaller than this cutoff threshold. This solution could potentially be extended by establishing different cutoff thresholds with respect to different classes. Correspondingly, at the inference phase, given a testing image, if its probability with respect to any class is smaller than the corresponding cutoff threshold, it then would be considered to be an outlier.
>
> Clearly, interesting future work, but beyond the scope of this current project. We will thus point at this idea as future work in Section 7.

---

> ### Comment · AnonReviewer2 · 2018-11-29
> **Response to the rebuttal**
>
> Thank you for the detailed rebuttal. This has been extremely helpful in better understanding the problem setup and the proposed scheme.
>
> My question regarding "supervised setting" needs clarification -- I was asking whether the proposed scheme can be used in an unsupervised setting where there are no labels what so ever and the goal is to find outliers, not a scenario where the model/scheme is presented with labeled outliers. Upon better understanding the scope of this paper, the problem being considered (and subsequently the scheme proposed to solve this problem) is closely tied to general supervised learning -- the problem is to identify/flag test points that do not belong to any of the classes present in the training set (and hence the trained model probably tries to shove the test point into one of the known classes).
>
> But this problem setup further complicates my understanding regarding the choice of k. The assumption is that all training samples are inliers. Outlier detection is needed during testing/inference to identify/flag images from potentially unknown classes. In that case, the threshold parameter k should really be k=0. Any k > 0 seems unjustified. We are essentially saying that k of the inliers are now outliers (for no good reason to the best of my understanding). Moreover, making 5000 inlier training points outliers implies that (potentially) a non-trivial number of test images might get spuriously flagged. Does the testing accuracy reported in the Tables 1 & 2 consider the spuriously flagged test points as mistakes? Or is the testing accuracy being computed without regard to the outlier flagging process?
>
> The problem setup also makes me feel that the empirical evaluation scheme (training on one dataset, testing for outliers on a completely different dataset) seems somewhat weird -- the problem scenario used as motivating example implies that the testing image is still probably from the same domain, but the image is from a class not present in the training set. Then an appropriate experiment would be something like training MNIST on digits 1-9 and testing digit 0 for outlier detection and testing digits 1-9 for testing accuracy (and counting images spuriously flagged as unknown class as mistakes).
>
> On a related note, for models trained on a particular dataset (say MNIST), how are the images from different datasets (say CIFAR-10) with different shape/dimensions input to the model for inference? After the convolution layers, wouldn't there be a mismatch between the number of features created and the expected input size to the FC layer? Are the images just cropped to match the shape of the training images?
>
> Thank you for the new experiments regarding choice of k and use of isolation forest. These are very informative. My proposal was to use the features right before the softmax layer, while the presented experiments in the appendix appear to use the output of the convolution layers. It looks like the deep neural forest works on features from the FC layer. So is there a reason why the isolation forest is only used on the convolution features of the trained model? Or have I misunderstood the experimental setups?

---

> > ### Author Response · Authors · 2018-12-08
> > **Response to the new comments from Reviewer 2 (Part 1)**
> >
> > We thank the reviewer for these additional questions and suggestions; and below we briefly summarize the questions we respond to along with our response.
> >
> > [REVIEW: Choice of parameter k]: It seems the assumption is that all training samples are inliers. Outlier detection is needed during testing/inference to flag images from potentially unknown classes. In that case, the threshold parameter k should really be k=0. Any k > 0 seems unjustified. You are essentially saying that k of the inliers in training set are now outliers (for no good reason to the best of my understanding).
> >
> > RESPONSE: Our observation here is that indeed there are outliers in the training imagery data set. One type of these outliers in the training set are mislabeled images. As an example, there are mislabeled images even in MNIST in spite of it being a widely adopted  benchmark dataset for image classification. We posted several example images of mislabeled data from MNIST at an anonymous URL https://drive.google.com/file/d/1XcCMhGCKS32sPJL4gq_9cvb61G-lySzv/view?usp=sharing where T denotes the label of the image.
> >
> > Leveraging this observation, our approach uses the parameter k to establish an outlierness cutoff threshold ct at training time, which is then used to detect outliers at testing time. The reason is that it is hard for the users to explicitly set an outlierness cutoff threshold ct, while it is more intuitive for the users to set this parameter k.
> >
> > Note in addition to identifying the testing images that are from unknown classes during testing phase, as a by product, our IOD approach also help find mislabeled images in the training imagery. Since the mislabeled images in the training set could confuse the classifier, capturing and cleaning these outliers may increase the accuracy of the classifier.
> >
> > [REVIEW: Measure of the testing accuracy reported in Tables 1, 2]: Making 5000 points in CIFAR-10 training points as outliers implies that (potentially) a non-trivial number of CIFAR-10 testing images might get spuriously flagged as outliers. Does the testing (classification) accuracy reported in the Tables 1 & 2 consider the spuriously flagged testing points as mistakes? Or is the testing (classification) accuracy being computed without regard to the outlier flagging process?
> >
> > RESPONSE: We thank the reviewer for this good question. It is true that using our IOD approach, some images in the CIFAR-10/MNIST testing set will be flagged as outliers by the model trained on the CIFAR-10/MNIST training images even if k is set much smaller than 5000, while the original image classifier will assign them to one of the target classes. Our results on the testing (classification) accuracy reported in Tables 1 and 2 indeed have counted these images as testing (classification) errors.
> >
> > Counting these images as classification errors only affects the classification accuracy of our approach to a limited degree. The reason is that many of these testing images tend to be either mislabeled or indeed look quite different from the majority of the images in their corresponding labeled classes. Hence they tend to be mis-classified by the classical CNN classifier that does not have the reject function. Based on our evaluation, our IOD method flags 51 MNIST testing images and 871 CIFAR-10 testing images as outliers, out of which 29 MNIST testing images and 657 CIFAR-10 testing images are mis-classified by the classical CNN classifier. Therefore, the increase in classification error on MNIST/CIFAR-10 goes up by only 0.22%/2.14% due to the reject function.

---

> > ### Author Response · Authors · 2018-12-08
> > **Response to the new comments from Reviewer 2 (Part 2)**
> >
> > [REVIEW: Empirical evaluation scheme]: The problem scenario used as motivating example implies that the testing image is still probably from the same domain, but the image is from a class not present in the training set. Then an appropriate experiment would be something like training MNIST on digits 1-9 and testing digit 0 for outlier detection and testing digits 1-9 for testing accuracy (and counting images spuriously flagged as unknown class as mistakes).
> >
> > RESPONSE: We agree that the experiment that the reviewer suggests concerning staying within the same domain sounds reasonable. It in fact corresponds to the type of experiments we had already conducted before. Namely, our previous CIFAR-10 experiments (see Sec. 5.2) which train a model on CIFAR-10 data and then test this model on CIFAR-100 data for outlier detection fall under this category, since CIFAR-10 AND CIFAR-100 are from the same source. These experiments confirm the effectiveness of our approach in detecting outliers from the same domain.
> >
> > However, based on the reviewer's suggestion, we now run additional experiments by training MNIST on digits 1-9 and then testing digit 0 for outlier detection. Our results show that our IOD-based approach (IOD-max-route) detects 93.2% digit 0 images as outliers, while the alternative methods, namely weighted-sum, Deep-SVDD, and AnoGAN only detect 50.59%, 62.09% and 4.5% digit 0 images as outliers respectively under the same experimental setting. In addition, the classification accuracy of our approach is still above 99%.
> >
> > We will add these new results to our manuscript, as soon as we are permitted to update our ICLR manuscript during this review process.
> >
> > [REVIEW: Dataset pre-processing:] For models trained on a particular dataset (say MNIST), how are the images from different datasets (say CIFAR-10) with different shape/dimensions input to the model for inference? After the convolution layers, wouldn't there be a mismatch between the number of features created and the expected input size to the FC layer? Are the images just cropped to match the size of the training images?
> >
> > RESPONSE: We either cropped or enhanced the testing images to match the size of the training images. More specifically, when testing CIFAR-10, CIFAR-100, and SVHN datasets on the model trained for MNIST, we change the image to gray scale and take central crops of the images. On the other hand, when testing MNIST on models trained for other datasets, we add zero padding on each border of the image and increase its color channel from 1 to 3 by copying the original gray image 3 times.
> >
> > These have also been described in our experimental section (Parameter Settings, Sec. 5.1).

---

> > ### Author Response · Authors · 2018-12-08
> > **Response to the new review from Reviewer 2 (Part 3)**
> >
> > [REVIEWER: New experiments of varying k]: The proposal was to use the features right before the softmax layer as input to Isolation Forest, while the presented experiments in the appendix appear to use the output of the convolution layers. It looks like the deep neural forest works on features from the FC layer. So is there a reason why the isolation forest is only used on the convolution features of the trained model?
> >
> > RESPONSE: We had understood the suggestion to be to consider using the input of the softmax layer as input features of Isolation Forest. In fact we also tested a network design where the Isolation Forest used the input of the softmax layer as feature. The results were extremely poor -- lower than 1% in all cases. Hence, we did not present these results in our previous response. The reason for this poor performance is that the input of the softmax layer corresponds to the weighed sum computed by the final FC layer. As described in our paper at the beginning of Sec. 2, given one image, a weighed sum is computed with respect to each target class. It is then transformed by the softmax function to a probability from 0 -- 1. The image is then assigned to the class with the largest probability. So this weighed sum can be interpreted as relative measurement of how likely it is that the image falls into one target class. It is however not necessarily effective in representing the key features of an image.
> >
> > In the Isolation Forest related experimental study, we reported the results produced by using the output of the convolution layers as input features to Isolation Forest, since to the best of our understanding, in deep neural networks typically the intermediate states at the convolutional layers are considered as features [1]. This works much better than using the input of the softmax layer as features to the Isolation Forest.
> >
> > We would like to clarify that the deep neural forest (either the original deep neural forest as well as our modified version) in fact also use the output of the convolution layer as input features to the decision forest. However, since the number of nodes at the convolutional layer (the number of features) may not match the number of nodes in the decision forest, the deep neural forest needs a single FC layer to connect the neural network to the decision forest. We apologize, as this confusion may have been caused by our Figure 2. We will thus modify this figure in the future revision of the manuscript to make this point more explicit.
> >
> > [1] Visualizing and Understanding Convolutional Networks, ECCV2014

---

### Official Review · AnonReviewer3 · 2018-11-02
**outlier detection using decision forest**

**Rating:** 5
**Confidence:** 3

**Review:**

The paper proposes a decision forest based method for outlier detection and claims that it is better than current methods.

A few questions:
What is the precise definition of maximum weighted sum? Why not using maximum probability instead in Figure 1? Are they equivalent? What does this 8.1701 threshold refer to? What architectures you use for the experiment in Section 2?

Comments:
The observation that simple methods for outlier detection are not good enough is interesting, and deserves deeper understanding.
However, directly calculating max. prob. may be a weak baseline. A stronger method to compare with would be using dropout during testing, see [1], which is easy to calculate and very practical (can easily be deployed to other tasks such as sequence tagging).
The extensibility of the proposed method is not clear to me.

Also, the reason that the observed failure of detection happens may due to the optimization procedure, i.e., how you train the model matters. The authors should provide the details of the training methods and architectures, along with the observation.

The baseline compared in the experiments are methods that do not use the classification feature. It would be necessary to compare with stronger baselines, such as using dropout.

Typo:
'a sample x $\in$ based on its features'

Reference:
[1] Dropout as a Bayesian Approximation: Representing Model Uncertainty in Deep Learning, by Yarin Gal, Zoubin Ghahramani

---

> ### Author Response · Authors · 2018-11-26
> **Response to Reviewer 3 (Part 1): the comments on details**
>
>
> [COMMMENT FROM REVIEWER]: (1) the definition of maximum weighted sum; (2) Why not using maximum probability in Figure 1? are they equivalent? (3) What the 8.1701 threshold refer to; (4) the architectures used for the experiment in Section 2
>
> RESPONSE:
> Thank you for the detailed review. We have carefully revised Section 2 of our paper to include our responses to your questions, as explained below. The code and models used in this work will be made public  after the double blind review process is completed.
>
> (1) The maximum weighted sum corresponds to the input of the softmax layer. For each class $C_i$, the final fully connected layer before the softmax layer produces a weighted sum score $s_i$. The score is computed by sum(F_j w_{ji} | 0<j<n+1)  where $F_j$ is a feature and $w_{ji}$ is the learned weight of the FC layer that connects $F_j$ and class $C_i$. The maximum weighted sum is the largest weighted sum score among all classes, defined as max(s_1,s_2, ..., s_m).
>
> (2) The maximum weighted sum score is not equal to the maximum probability. Probabilities can be computed by applying a softmax layer to the weighted sum scores. That is softmax(s_1,s_2,...s_m) = (p_1,p_2,...p_m).  The maximum probability then corresponds to the largest probability defined as max(p_1,p_2,...,p_m). In other words, the maximum weighted sum score  corresponds to the maximum score before the softmax layer, while the maximum probability is the maximum score after the softmax layer.
>
> We also worked with  this maximum probability as the outlierness measure of each image. However, we discovered that using maximum probability performed worse than using maximum weighted sum. Therefore, we selected the better of the two, namely,  the maximum weighted sum as our baseline method.
>
> (3) As explained in Sec. 2, the constant "8.1701" shown in Figure 1 is the cutoff threshold used in detecting outliers. It corresponds to the 5000-th smallest maximum weighted sum among the images in the training set CIFAR-10. Then at the inference time we consider images with maximum weighted sum smaller than 8.1701 as outliers. This cutoff threshold is variable.
>
> (4) The architecture we used for the experiments in Section 2 is identical to the ones used in our experimental section (Section 5. 2). The architecture is similar to the VGG-13 model with Batch Normalization. Specifically, the number of channels for the convolutional layers are [32, 32, ’M’, 64, 64, ’M’, 128, 128, ’M’, 256, 256, ’M’, 128, 128, ’M’], where ‘M’ is the max-pooling layer with kernel size=2, stride size=2. The kernel size for each convolutional layer is 3. Batch normalization and Relu functions are applied after each convolutional layer. We previously had already described the training process we used in detail in our experimental section.

---

> ### Author Response · Authors · 2018-11-26
> **Response to Reviewer 3 (Part 2): the comparison to the Dropout method**
>
>
> [COMMMENT FROM REVIEWER]:  Compare with the method that uses dropout during testing [1]
>
> RESPONSE:
>
> Response: We thank the reviewer for pointing us to [1] as an additional baseline. We have now followed your suggestion and also compare against this method. The results are worse than the previous baseline used in our paper. The results are detailed below and also incorporated into our revised paper (Appendix D ).
>
> This dropout paper [1] proposed a new theoretical framework casting dropout used in training deep neural networks as approximate Bayesian inference in deep Gaussian processes. A direct result of this theory gives us a tool to model uncertainty using dropout in NNs. As discussed in paper [1], dropout is applied during  inference. Given an image, we can get a range of softmax input values for each class by measuring 100 stochastic forward passes of the softmax input. If the range of the predicted class intersects that of other classes, then even though the softmax output is arbitrarily high (as much as 1 if the mean is far from the means of the other classes), the uncertainty of the softmax output can be as large as the entire space. In other words, the size of intersection signifies the model’s uncertainty in its softmax output value -- i.e., in the prediction. The larger the intersection is, the more uncertain the model is in its prediction.
>
> So we agree that this uncertainty could be used as a score to measure the outlierness of an image. Therefore, as suggested by the reviewer, we evaluated this dropout method proposed in [1] as a new baseline method.
>
> In the added experiments, we use the identical network architecture suggested in the github repository for [1] (https://github.com/yaringal/DropoutUncertaintyCaffeModels/tree/master/cifar10_uncertainty). When the method is applied on the CIFAR-10 training data, for each CIFAR-10 image, we forward it to the model 100 times and record the softmax input values for each class. The "outlierness" (uncertainty) of the image is defined as the intersection between predicted class and all other classes. We use the 5000-th largest value in the training as the cutoff threshold. Specifically, if an image has an uncertainty larger than the threshold, it is considered to be as an outlier. Then we forward each CIFAR-100 image 100 times in the model and record the outlier score for each CIFAR-100 images. Unfortunately as shown in Appendix D of our revised paper, the accuracy of this outlier detection scheme is only ~53%. That is, this is even worse than our maximum weighted sum baseline which has an accuracy of ~70%.
>
> We also applied the Dropout method on MNIST training images. Again we use the network architecture suggested by the authors in their repository. We forward each MNIST image 100 times in the model and compute the outlierness score. However, when we use the 200-th largest outlierness score in MNIST training as the outlierness cutoff threshold (the same parameter setting to our MNIST experiment in Appendix C), the accuracy in detecting CIFAR-10 images (also forwarded 100 times in the model) as outliers is lower than 10%. When we increase the parameter from 200 to 5000, its accuracy in detecting outliers increases to 48.13%, which is much much lower than our proposed method (above 90%), although clearly in this case the parameter setting biases towards the Dropout method.

---

> ### Author Response · Authors · 2018-11-26
> **Response to Reviewer 3 (Part 3): the extensibility of the proposed method**
>
>
> [COMMMENT FROM REVIEWER]: The extensibility of the proposed method.
>
> RESPONSE: We are not entirely certain what the reviewer means by extensibility. We are guessing here that the reviewer may be asking us about applicability to other applications and data sets. To address this,  we would like to point out that our approach is broadly applicable in a rich variety of real world applications for two reasons:
>
> (1) It resolves a significant limitation of  traditional image classifiers. Given one testing image, an existing CNN image classifier will assign this image to one of the classes observed in the training set, even if it does not belong to any known class in the training data set. For example, given a cat image, if we test it on a CNN model trained using MNIST, this cat image will be erroneously assigned to one of the digit classes. In the real applications, it is common for images supplied at inference time to not belong to any class known in the training data  -- for example, consider an autonomous vehicle trained mostly on urban imagery taken to the desert, where it sees sand, cacti, and tumbleweed for the first time. Our approach thus enhances any of the existing CNN-based classifiers with this powerful "rejection" ability. That is, it no longer blindly assigns a testing image to one of the known classes. Instead, an image will be rejected as being an outlier if it does not "sufficiently" belong to any of the existing classes.
>
> (2) Real applications tend to have a sufficient amount of normal data, and thus are able to more easily provide us with a large amount of labeled normal data for training the classification model, while they lack access to labeled outliers due to the rarity of outliers. Thus, an approach, such as ours, that uses only labeled inliers, and does NOT rely on the availability of outlier labels is a preferred situation in practice.

---

### Official Review · AnonReviewer1 · 2018-11-08
**Review of submission 1492**

**Rating:** 4
**Confidence:** 4

**Review:**

Summary: This paper modifies an existing technique designed for image classification to make it applicable to outlier detection.


Strengths: The outlined problem is of significant practical importance.

Weaknesses:
- The improvement over the existing method is incremental;
- The regularization on routing decision may not really be necessary as, in DNDF, the soft splits start as uniform and gradually converge to something close to hard splits; this is discussed in the supplementary material of the DNDF paper;
- the datasets tested are standard image datasets, not even captured from vehicles or video surveillance. The SVHN (street view numbers) dataset is the closest the experiments get to the motivating application.
Overall assessment: reject

Recommendations for the authors: Test on a surveillance or street view benchmark. Even then, it's questionable whether the paper is suitable for ICLR due to lack of methodological novelty.


Note: I'd like to apologize to the authors for the delay in submitting this review. It was due to a technical error on my part (I thought the reviews had posted, but they had not). In the spirit of independent evaluation, this review was not influenced by the other comments on this paper. I will follow-up with a response which will take into account the existing dialogue.

---

> ### Author Response · Authors · 2018-11-26
> **Response to Reviewer 1: an effective solution to an extremely important problem**
>
>
> [COMMMENT FROM REVIEWER]: The improvement over the existing method (DNDF) is incremental
>
> RESPONSE: Our approach represents an effective solution to an extremely important problem. In fact, our approach significantly outperforms the state-of-the-art in the accuracy of outlier detection as shown in our experimental study. While our approach leverages some of the DNDF principles, we introduce several critical insights to render it effective at detecting outliers.
>
> First, given that the DNDF approach was designed for improving classification accuracy, it was not obvious it would be applicable to tackling the outlier detection problem. Indeed, we are the first to leverage DNDF for addressing the outlier detection problem. We do this by unifying the best practice of unsupervised outlier detection with our observation that the max route of each tree in DNDF effectively captures the outlierness of each image.
>
> Second, we refine the core method with several technical innovations to assure effective outlier detection, while concurrently also yielding high accuracy for image classification. By this, the existing CNN-based image classifiers are enhanced to have the ability to reject a testing image as being an outlier if it does not ``sufficiently'' belong to any of the existing classes known in the training data.
>
> In particular, we proposed two new techniques, namely, an information-theoretic regularization strategy based on routing decisions and a new network architecture that ensures that each tree in the forest is completely independent. Further, as additional innovation, we designed a new joint learning method that optimizes the parameters for the decision node and for the prediction nodes in one integrated step through back-propagation, abandoning the two-step optimization strategy used in DNDF.
>
> Based on the above described innovations, our approach is not only technically novel but also useful, as it is highly effective at detecting outliers.
>
> We have revised the Proposed Approach and Contributions of the Introduction section (Section 1) to reflect the above discussion.
>
> [COMMMENT FROM REVIEWER]: The regularization on routing decision may not really be necessary as, in DNDF, the soft splits start as uniform and gradually converge to something close to hard splits.
>
> OUR RESPONSE: Indeed, we have had explored this question. Namely, although the original DNDF method does start from uniform soft splits and gradually converges to close to hard splits, directly using the max-route probability in DNDF to detect outliers does not achieve very high accuracy as we have demonstrated in our experimental study (Table 1).
>
> Our proposed regularization strategy based on routing decisions, namely penalizing large entropy probability distributions of the routing decision, significantly outperforms this original DNDF design for the problem of detecting outliers.
>
> As illustrated in Table 1 of our experimental section, using our regularization method, the accuracy of detecting CIFAR-100 outliers out of the CIFAR-10 dataset increases from 84.64% to 94.69%, that is, we gain an over 10% improvement.  Further, the accuracy of detecting MNIST outliers increases from 67.59% to 94.94% representing a close to 30% improvement in accuracy.
>
> These experimental findings on benchmark data sets provide strong evidence that our proposed design, in particular with the addition of the proposed regularization strategy, is both necessary and highly effective.
>
> [COMMMENT FROM REVIEWER]: The datasets tested are standard image datasets, not even captured from vehicles or video surveillance.
>
> OUR RESPONSE: The datasets we used in our experiments are indeed commonly used in state-of-the-art image outlier detection papers such as [1] and [2]. In our experiments we focused on these datasets because this ensures a fair comparison of our proposed outlier detection approach against the state-of-the-art.
>
> In the future, we will be happy to extend our evaluation to surveillance or street view data sets -- especially if we can gain access to data sets labeled with outliers.
>
> We thus thank you for this suggestion.
>
> [1] Ruff, Lukas, et al. "Deep one-class classification." International Conference on Machine Learning. 2018
>
> [2] Zhou, Chong, and Randy C. Paffenroth. "Anomaly detection with robust deep autoencoders." Proceedings of the 23rd ACM SIGKDD International Conference on Knowledge Discovery and Data Mining. ACM, 2017.

---

### Public Comment · ~Andrey_Malinin1 · 2018-10-10
**Related work**

Hello! :) Very interesting work. You may find our work on predictive uncertainty estimation to be relevant relevant.

https://arxiv.org/pdf/1802.10501.pdf

---

### Meta-Review · Area_Chair1 · 2018-12-12
**Incremental contribution**

**Confidence:** 4
**Recommendation:** Reject

**Metareview:**

The paper proposes a decision forest based method for outlier detection.

The reviewers and AC note the improvement over the existing method is incremental.

Although  the problem is of significant practical importance, AC decided that the authors should do more works to attract the attention of a broader range of ICLR audience.